# Bisphosphonate-Related Atypical Femoral Fractures in Patients with Autoimmune Disease Treated with Glucocorticoids: Surgical Results for 20 Limbs

**DOI:** 10.3390/jcm13041027

**Published:** 2024-02-10

**Authors:** Tomofumi Nishino, Kojiro Hyodo, Yukei Matsumoto, Yohei Yanagisawa, Masashi Yamazaki

**Affiliations:** Department of Orthopaedic Surgery, Institute of Medicine, University of Tsukuba, 1-1-1 Tennodai, Tsukuba 305-8575, Ibaraki, Japan; pjxgr965@tsukuba-seikei.jp (K.H.); yukeimatsumoto@tsukuba-seikei.jp (Y.M.); yanagisawa@tsukuba-seikei.jp (Y.Y.); masashiy@tsukuba-seikei.jp (M.Y.)

**Keywords:** atypical femoral fracture (AFF), bisphosphonate (BP), autoimmune disease, glucocorticoid (GC)

## Abstract

Background: Glucocorticoids induce osteoporosis, while bisphosphonates treat it, yet both can lead to atypical femoral fractures. Patients on both agents may face challenges in healing from such fractures due to their pathophysiology and pharmacological effects. Methods: Intramedullary nail surgery was performed on 20 limbs in 19 patients with atypical femoral fractures and autoimmune diseases, who had received bisphosphonates for GC-induced osteoporosis. The average durations of glucocorticoid and bisphosphonate use were 17 and 9 years (standard deviation: 7.59 and 4.35), respectively, and the mean follow-up period was 66 months. Fifteen and five limbs were fractured at the subtrochanter and diaphysis, respectively. The surgical techniques (type of nail) and additional procedures performed in these cases were examined. The post-operative alignment and reduction status on radiographs were examined to determine their relationship with post-operative outcomes. Results: Cephalomedullary long nails were inserted in nine limbs and antegrade intramedullary nails in 11 limbs. As an additional surgical procedure, open reduction, bone grafting and drilling were carried out on six, two, and five limbs, respectively. Regarding malalignment on radiographs, AP images showed varus in four limbs, and lateral images showed extension in two limbs. Regarding the cortical discontinuity, the distal fragment of the 11th limb shifted posteriorly in the lateral view. Gaps at the fracture sites were observed in 11 limbs. As a result, bone union was confirmed in 13 limbs. Five of the seven nonunion limbs required additional surgery. When comparing union and nonunion, open reduction and drilling were involved in nonunion limbs. Conclusion: The surgical outcomes of atypical femoral fractures in patients with autoimmune disease and on long-term glucocorticoids and bisphosphonates were poor. Although it is not possible to affirm for sure based on these results alone, management with prophylactic surgery before complete fracture is considered to be required to improve outcomes.

## 1. Introduction

Glucocorticoids (GC) are widely used for the treatment of various inflammatory, immunological and allergic diseases. Osteoporosis is one of the main complications of long-term GC administration, and the resulting fragility fractures not only reduce activities of daily life but also directly affect life expectancy [1,2]. Bisphosphonate (BP) is the first choice for the prevention and treatment of GC-induced osteoporosis and its use is recommended in Japanese guidelines [3]. BPs are widely utilized in the treatment of bone disorders such as osteoporosis and bone metastases. These agents exert their effects by inhibiting bone resorption and improving bone density, thereby enhancing bone strength. The primary mechanisms of action of BPs are as follows: Inhibition of bone resorption: BPs reduce bone resorption by suppressing the activity of osteoclasts, the cells responsible for bone breakdown. This inhibition mitigates bone destruction and prevents a decline in bone density.

Promotion of bone formation: BPs have also been suggested to increase the activity of osteoblasts, the bone-forming cells. This promotes bone regeneration and repair, contributing to improvements in bone density.

Other effects on bone quality include the following.

Bone remodeling: BPs influence the formation and maintenance of bone remodeling units, which play a crucial role in bone repair and regeneration. The promotion of bone remodeling by BPs contributes to overall bone health.

Repair of microdamage: BPs may play a role in the repair process of microdamage within bones, including stress fractures. This contributes to the enhancement of bone strength and durability.

Alleviation of pain: In conditions such as bone metastases, BPs alleviate pain by suppressing bone resorption and reducing the destruction of bone tissue.

These mechanisms collectively underscore the efficacy of bisphosphonates in the treatment of various bone disorders, highlighting their multifaceted impact on bone health and quality.

In contrast, BP exerts its effect by suppressing osteoclasts, and it has recently been suggested that the long-term use of BP can induce a condition called severely suppressed bone turnover [4], resulting in atypical femoral fracture (AFF) [5,6]. The term “severely suppressed bone turnover” is a phenomenon particularly highlighted in association with the use of BPs. It denotes a condition in which BPs excessively inhibit bone resorption, leading to a diminished capacity for bone regeneration and repair processes. While BPs primarily improve bone density by suppressing bone resorption, this effect can sometimes become excessive, hindering normal physiological bone turnover. Recognized as “severely suppressed bone turnover”, this state poses risks to bone strength and health, as it diminishes the natural processes of bone regeneration and repair. The prolonged or high-dose use of BPs may precipitate this condition, increasing the risk of delayed fracture healing or bone abnormalities. In extreme cases, serious adverse effects such as jaw osteonecrosis or atypical femoral fractures may occur. To minimize such risks, physicians must tailor the dosage and duration of BP therapy to each patient’s individual circumstances. Additionally, the regular monitoring of bone density and prompt identification and management of side effects during bisphosphonate therapy are essential. Owing to the pathophysiology of AFF, it takes time for bone union to occur even after the BP is withdrawn, and there are many cases with complications of delayed healing and pseudoarthrosis. GC interferes with fracture healing due to its chronic effects, which increase osteoclasts and suppress osteoblasts [7]. In the present study, we examined the results of surgical treatment of AFF, and the status of reduction of the fracture in a group of patients with autoimmune disease who had undergone long-term treatment with BP for GC-induced osteoporosis.

## 2. Materials and Methods

### 2.1. Inclusion and Exclusion Criteria

Since January 2009, there were 23 limbs from 22 cases with complete fractures that met the diagnostic criteria of the American Society for Bone and Mineral Research (ASBMR) 2nd Task Force [6], which were operated on at our hospital, and where follow-up was possible for at least 1 year. To satisfy the case definition of AFF, the fracture must be located along the femoral diaphysis from just distal to the lesser trochanter to just proximal to the supracondylar flare. In addition, at least four of five major features must be present. None of the minor features are required but have sometimes been associated with these fractures. The five major features are as follows: The fracture is associated with minimal or no trauma, as in a fall from a standing height or less. The fracture line originates at the lateral cortex and is substantially transverse in its orientation, although it may become oblique as it progresses medially across the femur. Complete fractures extend through both cortices and may be associated with a medial spike; incomplete fractures involve only the lateral cortex. The fracture is noncomminuted or minimally comminuted. Localized periosteal or endosteal thickening of the lateral cortex is present at the fracture site (“beaking” or “flaring”). The four minor features are the following: Generalized increase in cortical thickness of the femoral diaphysis. Unilateral or bilateral prodromal symptoms such as dull or aching pain in the groin or thigh. Bilateral incomplete or complete femoral diaphysis fractures. Delayed fracture healing.

In order to understand the relationship with autoimmune diseases treated by GC in the current study, it was the policy to exclude cases that did not fit this description from the study. Therefore, we decided to exclude three limbs of three cases. Finally, 20 limbs of 19 cases with autoimmune diseases were included (Table 1). 

### 2.2. Patient Demographics

In all cases, BP was administered as a prophylaxis and treatment for GC-induced osteoporosis. The average age at the time of injury was 64 (41–85) years, and all but one patient was female. The mean follow-up period was 66 (15–144) months, including two cases with three limbs that died (#2, 7 and 8) and two cases with two limbs that were transferred (#1 and 4) during the course of this study. The average body mass index was 25.2 (18.7–29.4) kg/m^2^. Of the 21 limbs, 8 were right, 12 were left, and 1 case (#7 and 8) had bilateral simultaneous complete fractures requiring surgery on both sides. To determine the fracture site, a subtrochanteric femoral fracture was defined as the fracture line present in the proximal femur at intervals from the lesser trochanter to the ischium. The main fracture line included the inferior margin of the lesser trochanter and the boundary between the proximal femur and the proximal third of the diaphysis (approximately 5 cm from the inferior margin of the lesser trochanter). The results showed that the fracture sites were at the subtrochanter in 15 limbs and the diaphysis in 5 limbs. Sixteen limbs had fracture lesions on the contralateral side, and one limb (#5) had already been completely fractured and an intramedullary nail was inserted at another hospital. There were 14 cases of prodromal symptoms such as thigh or groin pain in 15 limbs. Rheumatoid arthritis was the most common disease treated with GC (nine limbs of nine cases), followed by systemic lupus erythematosus (five limbs of five cases), adult Still’s disease (three limbs of two cases), myasthenia gravis, idiopathic interstitial pneumonia (two limbs of two cases), IgG4-related diseases, dermatomyositis, and polymyositis (duplicated in four cases). The average length of time on GC before injury was 17 (7–34) years (standard deviation: 7.59). All patients had been taking internal BP preparations for an average of 9 (4–20) years (standard deviation: 4.35). Alendronate alone was the most common formulation (12 limbs of 11 cases), with 6 cases and 6 limbs in which the formulation was switched in the middle of the course. Alendronate was administered in 17 cases (18 limbs) during the study period and was the most frequent, because alendronate has been used in Japan for a long time because it was approved and marketed for the treatment of osteoporosis in the early 2000s. None of the patients showed femoral lateral bowing [8] on X-ray. Table 2 shows bone-related markers and bone density measured by dual-energy X-ray absorptiometry in this series. However, it was not possible to provide all the results; therefore, the data are presented only for reference.

### 2.3. Treatment Strategies

As a rule, a cephalomedullary long nail is indicated for subtrochanteric fractures and the proximal part of the femoral diaphysis, and an antegrade intramedullary nail is indicated for fractures in the central and distal parts of the femoral diaphysis. However, the actual fracture type and medullary cavity diameter of the actual case was also taken into consideration, and the type of nail was ultimately determined by the attending physician. Surgery was performed using a radiopaque fracture traction table, and anatomical reduction was attempted using various methods. Nails were generally inserted after at least 2 mm of over-reaming and fixed with lag screws from the neck to the femoral head or from the greater trochanter to the lesser trochanter, depending on the nail type. In cases where closed reduction could not be achieved, open reduction was used. In cases in which a gap in the fracture was evident, bone grafting was performed using reamed bone from the proximal femur. Additional procedures such as drill perforation were performed in cases with significant sclerotic images and thickening. Postoperatively, loading was initiated in stages, starting with partial loading. Low-intensity pulsed ultrasound was used as an adjuvant therapy from the fifth case onwards. In the postoperative period, BP was discontinued in all cases, and teriparatide was administered after the seventh case; however, if side effects appeared, teriparatide was discontinued. 

### 2.4. Assessment Methods

Follow-ups were conducted by the physician in charge at standard intervals of 1, 3, and 6 months, and 1 year. The treatment modalities, additional procedures, adjuvant therapy, reduction alignment, cortical continuity, fracture gap, corrective loss of realignment, time to healing, and nonunion were retrospectively recorded. Alignment was evaluated using Egol et al.’s evaluation [9], and poor alignment was defined as an angular dislocation of >10° in each direction. To determine the horizontal dislocation, we examined whether the cortex was in contact with the cortex on the front and side surfaces [10]. Continuity was defined as the overlap of each cortical fragment and a gap of 1 mm at the fracture site. If the cortex did not overlap, the direction of the dislocation was recorded. Corrective loss of realignment was defined as a change in the angle between the AP and lateral images of more than 2° between the immediate postoperative radiographs and the radiographs taken at least 6 months after surgery. Radiographic healing was defined as bridging across three or four cortices and/or the disappearance of the fracture line at a glance based on standard AP and lateral views. Pseudarthrosis cases were classified according to pseudarthrosis morphology [11]. Complications, such as postoperative infection, neuropathy, and pulmonary embolism, were investigated. Fracture healing was determined by the surgeon in charge (TN or YM). Informed consent was obtained from all patients authorizing the use and publication of the data. This retrospective study was approved by the Ethics Review Committee of our institution (approval Code: H27-041). 

### 2.5. Statistical Analysis

Patient factors (age, sex, body mass index, fracture site, bilateral, prodromal, comorbidity, duration of GC, and duration of BP) and treatment factors (open reduction, bone graft, drilling, low-intensity pulsed ultrasound, teriparatide, reduction status, cortical continuity, fracture gap, and corrective loss) were used to determine whether bone union was statistically significant. For all factors, all tests were analyzed in a univariate analysis to see the relationship with the outcome (presence or absence of bone union). The significance level was set at *p* = 0.05. All statistical analyses were performed using the Statcel 3 software (OMS, Saitama, Japan).

## 3. Results

The results are summarized in Table 3. Cephalomedullary long nails were inserted in nine limbs and antegrade intramedullary nails in eleven limbs. As an additional surgical procedure, open reduction was performed on six limbs, bone grafting on two limbs, and drilling on five limbs. Adjuvant therapies included low-intensity pulsed ultrasound and teriparatide. Low-intensity pulsed ultrasound was used in 16 limbs of 15 cases after #5, and teriparatide was used in all cases after #5, except in cases in which drug-induced side effects occurred (#5, 10, and 19).

Regarding malalignment on radiographs, AP images showed internal rotation in four limbs, and lateral images showed extension in two limbs. Regarding cortical discontinuity on radiography, the distal fragment of the 11th limb shifted posteriorly in the lateral view. Gaps at the fracture sites were observed in 11 limbs. There were two limbs in the AP image and one limb in the lateral image due to corrective loss of realignment. Bone union was confirmed in 13 limbs at the final follow-up. The average time to bone union was 13 (5–36) months. Of the seven limbs that did not achieve bony union by the last observation, two (#3 and 10) showed implant fractures. One limb (#3) underwent reinsertion of a different type of cephalomedullary long nail and finally achieved bony fusion. One limb (#10) was replaced by hemiarthroplasty using a hook plate. No bone union or stem-loosening was observed. One limb (#13, Figure 1) was inserted with varus and had corrective loss, and the gap at the fracture site was enlarged and painful. Nail exchange was performed 18 months after the initial surgery, and bone graft and plate fixation were combined to obtain bone union [12]. One limb (#6), a hypertrophic nonunion with corrective loss in two directions, underwent dynamization [13] to remove the distal locking screw 20 months after the initial surgery, and ultimately achieved bony fusion. One limb (#15) had atrophic nonunion and underwent dynamization after 31 months, followed by bony fusion. The reason for the long time to dynamization was that the patient had no symptoms at all and refused the surgical proposal. Patients with atrophic nonunion in two limbs (#7 and 18) had few symptoms and were treated conservatively.

A comparison of the patient and surgical factors between the bone union and nonunion groups showed that open reduction (*p* = 0.007) and drilling (*p* = 0.031) were associated with nonunion. Five of the six limbs that underwent open reduction were nonunion, and four of the five limbs that underwent drilling were nonunion. No complications affecting the postoperative course were observed.

## 4. Discussion

A large body of evidence has accumulated for the treatment of osteoporosis using BP since its advent. Currently, the drug is positioned as the first line of treatment in terms of its objective: fracture prevention. However, AFF remains an unresolved issue.

Although AFF was initially highlighted as a BP-related condition [14], the 2010 Task Force [5] did not include BP use in the major category of this disease but rather as one of the drugs listed as a possible cause in the minor category. Although drugs containing BPs were removed from the sub-items in the 2nd Task Force [6], their impact cannot be ruled out based on the results of case–control and cohort studies [15]. 

GC was included in the 2010 Task Force as a sub-item along with BP. The chronic action of GCs increased the number of osteoclasts, suppressed osteoblasts, and caused apoptosis in bone cells [7]. Although it is not known whether GC alone causes AFF, it suppresses the bone metabolism, and long-term use and high doses are risk factors [16,17,18]. The following mechanisms have been postulated for GCs with regard to bone metabolism: they inhibit osteoblast function and reduce their ability to form new bones. In other words, they inhibit bone formation. They also increase osteoclast activity. This results in the faster resorption of existing bone and reduced bone density. This means increased bone resorption. In addition, it may decrease the activity of chondrocytes. This may prevent cartilage repair at the fracture site. This interferes with the fracture repair mechanism. Considering all these mechanisms, it is easily conceivable with regard to the use of GCs that the fracture healing mechanisms do not function, resulting in prolonged healing and pseudoarthrosis. In contrast, BP has been proven to reduce the risk of GC-induced fractures by suppressing osteoclast activity [19]. Therefore, Japanese guidelines stipulate the use of BP (alendronate and risedronate are recommended as grade A) to prevent GC-induced osteoporosis [3]. 

The risk of severely suppressed bone metabolism and atypical femoral fractures may vary depending on the type of BP and the patient’s demographics and condition [5,6]. There are several types of BPs, including alendronate, risedronate, and zoledronic acid, with varying dosing intervals and routes of administration. Studies suggest that the risk of suppressed bone turnover and atypical femoral side effects may vary depending on the type of BP. Some studies indicate that it may be particularly high with long-term use of intravenous zoledronic acid. Patient demographics and underlying medical conditions may also influence the risk of side effects associated with BP therapy. The risk of side effects may be higher in older patients due to factors such as reduced renal function and altered bone metabolism. Patients with a history of osteoporotic fractures may be at higher risk of side effects. As mentioned at the outset, the concomitant use of certain drugs, such as glucocorticoids and proton pump inhibitors, may increase the risk of side effects associated with bisphosphonates. Healthcare providers should consider these factors when making treatment decisions and monitor patients closely for potential side effects.

In the present study, we treated 20 limbs in 19 cases of AFF that occurred in patients with autoimmune diseases such as rheumatoid arthritis, who were taking GC and BP as prophylaxis long-term. Although there have been numerous reports of BP-related AFF outcomes, there have been no scattered studies of autoimmune diseases treated with GC, as in the present study. However, it is difficult to generate a coherent report on the outcomes in such patients because, as mentioned earlier, both GC and BP may be risk factors for the development of AFF, and both the pathogenesis and pharmacological effects of GC have negative effects on bone fusion. This is expected to increase the difficulty of treatment for AFF. Seven of the twenty limbs (35%) failed to achieve bony union and five limbs required additional surgery, which is a very poor result. 

The selected hospital is a university hospital that provides advanced medical care and has a very active department of collagen medicine. As a result, many patients with autoimmune diseases come to the hospital. Therefore, this is a highly skewed group of AFF patients and differs from the multicenter patient population we have investigated in the past [8].

Regarding surgical procedures, our hospital uses nails according to the recommendations of the Task Force [5]. A systematic review by Koh et al. [20] recommended intramedullary nails for complete fractures. The surgical technique involved over-reaming of at least 2 mm during reaming. Whenever possible, a traction table was used intraoperatively to reduce and pull the fracture into place. However, in AFF, the medullary cavity is sclerotic and eccentric for guide-pin insertion and reaming, and reduction is often difficult. If there was difficulty in reducing the fracture, we performed additional open reduction, and bone grafting and drilling were performed as needed. Statistically, open reduction and drilling were correlated with nonunion, which was the opposite of the expected outcome. Intraoperative findings may have had a significant influence on the presence or absence of drilling. Intraoperative findings that may cause the surgeon to consider the need for a procedure include the condition of the healing and the condition of the bone (whether it is sclerotic or not). If these conditions are poor, they are likely to remain the factors most inhibiting bone union. In other words, the presence of these findings was considered to be due to a preconceived notion that the bone was originally unfavorable for bone union.

Six of the seven nonunion cases were subtrochanteric fractures, all of which were atrophic. In general, surgical outcomes are not always good for subtrochanteric fractures because of the deforming anatomical forces [21]. In the present case series, three quarters of the fracture sites were subtrochanteric. Although the number of patients was small and no statistical difference was found (*p* = 0.61), it was expected that femoral diaphysis would be less common due to the nature of the underlying disease and patients on GC, and that subtrochanteric patients would have a poorer outcome for the reasons given above. This was not taken into account in the present study, as the emphasis was on manipulation at the time of surgery. Therefore, the fracture site should probably have been considered separately. Although this series was limited to cases that could not be treated by closed reduction, we believe that the direction and amount of the initial dislocation may have had an impact on the outcome. We consider this a reflection of this series and a challenge for future research. Subtrochanteric fractures are also a factor leading to surgery after conservative treatment of AFF [22], and it is important to perform prophylactic surgery for subtrochanteric fractures before complete fractures occur, especially in complex conditions such as the present case [23].

In general, AFF requires a long period of time for bone fusion, and in a study of 41 limbs in 33 cases by Egol et al. the average duration of bone fusion was 8.3 months [9]. The 13 limbs in our study that were fused also took a long time to heal, with an average duration of 13 months (#19, Figure 2). Good reduction is the basis of fracture treatment, and in AFF, good repair also shortens the time to bone union by 3.7 months [9]. It has also been reported that cortical disruption, residual gaps, and poor anteroposterior lateralization were poor performance factors in surgery [24]. As mentioned above, various previous studies suggest that a better healing situation is minimally necessary to aid in the mechanism of bone union. However, our study did not show the same results. In our studies [10], we observed the state of the surgical reduction in detail. However, the status of the reduction was not reflected in the results. This may be due to the small number of cases, but it may also be necessary to consider additional factors such as fracture site and fracture type. 

Six of the seven nonunion limbs had atrophic-type nonunions with subtrochanteric fractures, which may require some form of adjuvant therapy from a biological perspective. However, this study found no superiority of teriparatide and low-intensity pulsed ultrasound with respect to bone healing. Numerous articles have reported the benefits of teriparatide, and a recently published systematic review and meta-analysis showed that teriparatide reduced the rate of prolonged union and nonunion and shortened the fracture healing time [25]. Although the case series [26] are scattered with respect to low-intensity pulsed ultrasound, further research is required. As described above, even in common BP-related AFF, the surgical outcome is more difficult than in normal fractures, and even more so in the present series. Although no conclusions can be drawn from this study alone, it is highly likely that prophylactic surgery is required before complete fracture, including not only commonly described subtrochanteric fractures but also diaphyseal fractures, in order to improve outcomes.

This study investigates the treatment outcomes of AFF in patients with autoimmune diseases who were on long-term GC and BP. The study aims to assess the status of fracture reduction and bone union following surgical treatment, and to explore factors influencing treatment outcomes. While glucocorticoids are known to interfere with bone metabolism and healing processes, bisphosphonates, which are commonly used to mitigate GC-induced osteoporosis, may paradoxically contribute to severely suppressed bone turnover, potentially increasing the risk of AFF. This juxtaposition underscores the complexity of managing bone health in patients with autoimmune diseases. The research highlights the challenges encountered in the surgical management of AFF in patients with autoimmune diseases. Despite anatomical reconstruction during surgery, the outcomes were poor, with a significant proportion of limbs failing to achieve bone union. This underscores the difficulty in treating AFF, especially in patients with underlying autoimmune conditions and prolonged exposure to medications that affect bone metabolism.

The first limitation of the present study is that although it is a case series, it is not a comparative study and therefore it is not clear how autoimmune diseases treated with GC affect the surgical outcome. At our institution, there were only three patients who did not meet the conditions of the current study during the time period covered, so a comparison could not be made. We would like to increase the number of cases and conduct comparative studies in the future. However, the pathological conditions and the pharmacological effects of GC suggest that a positive effect is unlikely.

The second limitation of this series is that the cases were from 13 years ago, and there were multiple surgeons; therefore, the surgical techniques and treatment strategies were inconsistent. Due to the retrospective nature of the study, there was no agreement on the indications and methods of detailed open reductions. Future comparative studies are needed to increase the number of cases and adjust for bias due to differences in methods.

## 5. Conclusions

The surgical outcomes of 20 limbs of 19 cases of AFF in patients with autoimmune diseases and on long-term GC and BP medications were poor. Biological treatment may be necessary in addition to anatomical reconstruction during surgery. Although it is not possible to affirm for sure based on these results alone, management with prophylactic surgery before complete fracture is considered to be required to improve outcomes.

## Figures and Tables

**Figure 1 jcm-13-01027-f001:**
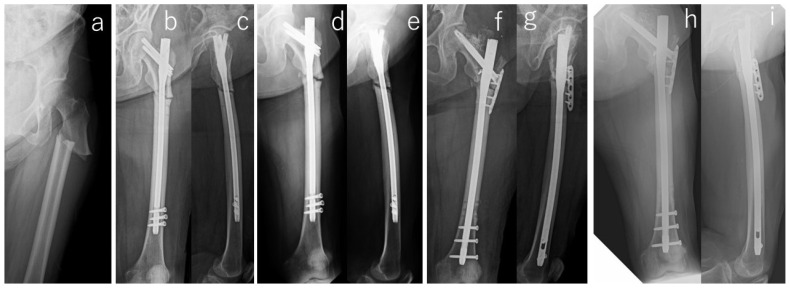
Radiographs of the right femur of #13. Preoperative radiography (**a**) showed transverse subtrochanteric fracture with lateral cortical thickness and breaking. Radiographs taken immediately after surgery showed residual varus malalignment (**b**) and fracture gaps (**b**,**c**). Radiographs at 18 months after surgery (**d**,**e**) showed progressive varus with corrective loss and enlarged fracture gaps. Revision surgery was performed (**f**,**g**) and ended in bone union (**h**,**i**).

**Figure 2 jcm-13-01027-f002:**
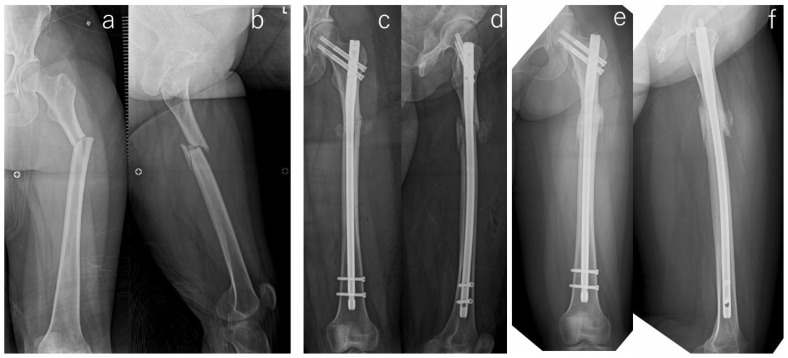
Radiographs of the right femur of #19. Preoperative radiographs (**a**,**b**) showed transverse subtrochanteric fracture with a distal third fragment. Radiographs taken immediately after the surgery showed residual varus malalignment (**c**) and the distal fragment translated posterior in the lateral view (**d**). Radiographs at five months after the surgery (**e**,**f**) showed bone union.

**Table 1 jcm-13-01027-t001:** Patient demographics.

#	Age (yr)	Gender	Body Mass Index (kg/m^2^)	Follow-Up Period (mo)	Affected Side	Location	Bilateral Lesion	Prodrome	Comorbities	Duration of GC Use (yrs)	Kind of BP	Duration of BP Use (yrs)
1	72	Female	29.4	60	Right	Subtrochanteric	+	+	Myasthenia gravis, Diabetes mellitus	28	Risedronate	5
2	54	Female	24.5	18	Right	Subtrochanteric	+	+	Dermatomyositis, Interstitial pneumonia	18	Alendronate	4
3	58	Female	23.4	146	Right	Subtrochanteric	+	-	Rheumatoid arthritis	12	Alendronate	6.5
4	54	Female	25.0	112	Left	Subtrochanteric	+	-	Rheumatoid arthritis	9	Alendronate	5.5
5	48	Female	29.4	127	Right	Femoral diaphysis	+	+	Systemic lupus erythematosus	21	Alendronate	10
6	67	Female	27.7	128	Left	Femoral diaphysis	-	-	Rheumatoid arthritis	30	Alendronate	5
7	78	Female	28.0	61	Right	Subtrochanteric	+	+	Adult Still’s disease	11	Alendronate	7
8	78	Female	28.0	61	Left	Subtrochanteric	+	+	Adult Still’s disease	11	Alendronate	7
9	50	Female	20.4	90	Left	Femoral diaphysis	+	+	Rheumatoid arthritis	20	Alendronate/Minodronate	6.5
10	79	Female	23.5	75	Left	Subtrochanteric	+	+	Adult Still’s disease	11	Etidronate/Alendronate	7
11	76	Male	18.7	77	Right	Subtrochanteric	-	+	Rheumatoid arthritis	14	Alendronate/Minodronate	7
12	67	Female	27.1	74	Left	Subtrochanteric	+	+	Interstitial pneumonia, Diabetes mellitus	9	Alendronate	9
13	73	Female	24.4	46	Left	Subtrochanteric	+	-	Systemic lupus erythematosus, Rheumatoid arthritis	20	Etidronate/Alendronate	12
14	58	Female	24.7	44	Left	Subtrochanteric	+	+	Systemic lupus erythematosus, Basedow’s disease	34	Alendronate	20
15	58	Female	29.1	43	Left	Subtrochanteric	+	+	Rheumatoid arthritis	19	Alendronate	19.5
16	56	Female	24.4	41	Right	Femoral diaphysis	+	-	IgG4-related disease	10	Alendronate	10
17	85	Female	27.4	39	Right	Femoral diaphysis	-	+	Myasthenia gravis	7	Alendronate	7.5
18	49	Female	20.6	34	Right	Subtrochanteric	+	+	Systemic lupus erythematosus, Rheumatoid arthritis	11	Minodronate	5.6
19	41	Female	26.3	18	Left	Subtrochanteric	+	+	Systemic lupus erythematosus, Glomerulonephritis	18	Alendronate/Minodronate	13
20	76	Female	21.4	15	Left	Subtrochanteric	-	+	Rheumatoid arthritis, Polymyositis	26	Alendronate/Minodronate	14.5

# 2, 7&8: death while f/u, # 7&8: simultaneous injury, # 5: operated already, # 1&4: change hospital while f/u.

**Table 2 jcm-13-01027-t002:** Bone-related markers and bone density.

	Bone-Related Markers: Unit (Normal Range)								Dual-Energy X-ray Absorptiometry	
#	Ca *	IP	ALP	BAP	Intact-P1NP	Total-P1NP	Urinary NTX	TRACP-5b	Lumbar Spine	Contralateral Femoral Neck
	mg/dL (8.5–10.5)	mg/dL (2.7–4.5)	U/I (104–338)	μg/L (3.8–22.6)	μg/L (27–109)	μg/L (18.1–74.1)	/mmol·Cre (14.3–89.0 )	mU/dL (120–420)	BMD (g/cm^2^)/T-Score/YAM (%)	BMD (g/cm^2^)/T-Score/YAM (%)
1	9.7	2.5	219	N.A.	N.A.	N.A.	30.9	N.A.	1.068/2.2/106	0.867/2.4/100
2	9.6	2.5	118	N.A.	N.A.	N.A.	N.A.	N.A.	N.A.	N.A.
3	9.1	3.4	218	10.9	N.A.	N.A.	45	N.A.	0.824/−1.7/81	0.718/−1.3/83
4	9.5	3.1	128	6.2	N.A.	N.A.	31.5	170	0.874/−1.2/86	0.700/−0.8/89
5	9.8	3.3	141	N.A.	35.3	N.A.	N.A.	162	0.987/−0.2/98	0.616/−1.6/78
6	9.3	4.3	335	N.A.	15.1	N.A.	N.A.	295	1.076/0.6/106	0.570/−2.0/72
7, 8	8.9	2.8	148	N.A.	12.6	N.A.	N.A.	147	1.209/1.8/120	N.A.
9	9.3	N.A.	248	N.A.	N.A.	11.8	N.A.	110	0.507/−4.6/49	0.476/−2.9/60
10	9.6	3.3	168	N.A.	N.A.	17.4	N.A.	321	0.705/−2.8/70	0.639/−1.4/81
11	9.1	3.2	122	N.A.	N.A.	11	N.A.	N.A.	0.841/−1.4/81	0.551/−2.5/64
12	9	3.6	174	N.A.	N.A.	13.7	N.A.	344	1.106/0.9/109	0.758/−0.3/96
13	9.3	3	207	8.9	N.A.	21.1	24.6	340	N.A.	N.A.
14	9.4	4	138	N.A.	N.A.	10.1	N.A.	231	1.067/0.5/106	N.A.
15	9.1	3.3	256	N.A.	N.A.	35.7	N.A.	665	1.156/1.3/114	0.885/0.9/112
16	8.9	3.2	184	N.A.	N.A.	10.7	N.A.	298	1.027/0.1/102	0.769/−0.2/98
17	9	4.1	286	N.A.	N.A.	23.8	N.A.	492	1.294/2.6/128	0.507/−2.6/64
18	8.9	4.7	27 *	N.A.	N.A.	11.7	N.A.	63	N.A.	N.A.
19	8.9	2.3	66 *	N.A.	N.A.	12.8	N.A.	170	0.841/−1.5/83	0.646/−1.3/82
20	9.2	3.6	69 *	N.A.	N.A.	67	N.A.	300	0.712/−2.7/70	0.402/−3.5/51

N.A.: Not available. * IFCC (International Federation of Clinical Chemistry) method, normal range: 38–113.

**Table 3 jcm-13-01027-t003:** Summary of the results.

Implant	Operative Procedure			Adjuvant Therapy		Malalignment		Cortical Discontinuity (-/Direction)		Fracture Gap	Correction Loss		Duration of Bone Union/Nonunion
	Open Reduction	Bone Graft	Drilling	LIPUS ^#^	Teriparatide	AP View	Lateral View	AP View	Lateral View		AP View	Lateral View	
Antegrade intramedullary nail	-	-	-	-	-	-	-	-	Posterior	+	-	-	6 months
Antegrade intramedullary nail	-	-	-	-	-	-	-	-	Posterior	+	-	-	10 months
Cephalomedullary long nail	-	-	-	-	-	Varus	-	-	Posterior	-	-	-	Non-union (atrophic) *
Cephalomedullary long nail	-	-	-	-	-	-	-	-	Posterior	-	-	-	16 months
Antegrade intramedullary nail	-	-	-	+	-	-	-	-	-	-	-	-	18 months
Antegrade intramedullary nail	-	-	-	+	-	-	-	-	Posterior	-	+	+	Non-union (Hyper) *
Antegrade intramedullary nail	+	-	+	+	+	-	-	-	Posterior	+	-	-	Non-union (atrophic)
Cephalomedullary long nail	-	-	-	+	+	-	-	-	Posterior	+	-	-	36 months
Antegrade intramedullary nail	-	-	-	+	+	-	-	-	Posterior	+	-	-	10 months
Cephalomedullary long nail	+	+	+	+	-	-	-	-	-	+	-	-	Non-union (atrophic) *
Cephalomedullary long nail	-	-	-	+	+	-	Extension	-	Posterior	+	-	-	21 months
Cephalomedullary long nail	-	+	+	+	+	Varus	-	-	-	-	-	-	6 months
Cephalomedullary long nail	+	-	-	+	+	-	-	-	-	+	+	-	Non-union (atrophic) *
Antegrade intramedullary nail	+	-	-	+	+	-	-	-	-	-	-	-	20 months
Cephalomedullary long nail	+	-	+	+	+	-	Extension	-	Posterior	+	-	-	Non-union (atrophic) *
Cephalomedullary long nail	-	-	-	+	+	-	-	-	-	-	-	-	6 months
Antegrade intramedullary nail	-	-	-	+	+	-	-	-	-	+	-	-	9 months
Antegrade intramedullary nail	+	-	+	+	+	-	-	-	-	-	-	-	Non-union (atrophic)
Antegrade intramedullary nail	-	-	-	+	-	Varus	-	-	Posterior	-	-	-	5 months
Antegrade intramedullary nail	-	-	-	+	+	Varus	-	-	-	+	-	-	8 months

^#^ Low Intensity Pulsed Ultra Sound. * Additional surgery required.

## Data Availability

The data are available from the corresponding author if required.

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
