# Peer review of "Bisphosphonate-Related Atypical Femoral Fractures in Patients with Autoimmune Disease Treated with Glucocorticoids: Surgical Results for 20 Limbs"

_jcm, 2024, doi:10.3390/jcm13041027_

Round 1
Reviewer 1 Report
Comments and Suggestions for Authors
introduction in the abstract should be refined , should describe main problem, state of art briefly, reason of the study
Materials in the abstract : please provide standard deviation for duration
Introduction : describe better the condition of severly suppressed bone disorder
describe current state of art of treatment based on current literature for the treatment of AFF and the aim of this study and why and what's interesting in this present study.
Why most patients got Alderonate and not zoldernic acid ?
why authors included different type of fracture ( subtroch and femoral diaphysis) as these fracture can vary on the degree of union / type of implant / risk of malunion (BP fracture usually occurs in subtrochanteric area)
Does LIPUS use improved union rate?
please provide ethical commite approval in the main text
does surgeries outcomes varied when senior surgeon who performed the surgery or resident ?
is there a role of direct adjunctive such as bone marrow aspirate or DBM ? , why bone grafting only used in some cases and not for others. what was the criteria to choose this approach ?
line 171 : why dynamization was done after 31 months ! why no treatment was done before !
how authors can explain that drilling was associated with higher rate of non union (while authors mentioned that drilling was done in case of sclerotic bone) which is mean that there is a confounding bias . i believe that the status of sclerotic bone which was the main cause of non union (not the drilling) . what was the confidance interval ? how different variables were elimianted ?
Comments on the Quality of English Language
english should be refined in better way. i recommend a native english speaker to correct minor errors
Author Response
Response to Reviewer 1
Thank you for taking the time in your busy schedule to review my paper. Thank you also for your many helpful and important comments and suggestions. I have prepared a new, highlighted version of the paper based on the comments of another reviewer and am sending it to you. Deletions have been bylined and new additions have been added in red. Please check it.
Also below is my response to your peer review comments. It is possible that there are some errors in my understanding, but I would be grateful if you would not hesitate to point them out and provide me with guidance. Thank you for your continued patience.
introduction in the abstract should be refined , should describe main problem, state of art briefly, reason of the study
→ I think you are right. It lacked clarity. The introduction to the abstract has been updated. (Line 23-28)
Materials in the abstract : please provide standard deviation for duration
→Thanks for your valuable remarks, the standard deviation of the duration of GC and BP was 7.59 and 4.35, respectively, and this information has been added to the abstract and text. (Line 31-32, 146-148)
Introduction : describe better the condition of severly suppressed bone disorder
→I thank you for pointing out a very important and basic matter. Another reviewer asked for a description of the basic mechanism of action of BP, to which I added a description of the mechanism of SSBT. (Line 57-78, 81-94)
describe current state of art of treatment based on current literature for the treatment of AFF and the aim of this study and why and what's interesting in this present study.
→I added a summary at the end of the discussion of the points you raised. (Line 350-362)
Why most patients got Alderonate and not zoldernic acid ?
→In Japan, the country of the current study, alendronate was first approved and marketed for the treatment of osteoporosis in 2002. The approval and marketing of zoledronic acid for the treatment of osteoporosis lagged far behind in 2016. There is a history of risedronate and minodronate being marketed more than a decade after alendronate was launched. As such, the majority of the products in this study are alendronate. We have added a description on this point. (Line151-154)
why authors included different type of fracture ( subtroch and femoral diaphysis) as these fracture can vary on the degree of union / type of implant / risk of malunion (BP fracture usually occurs in subtrochanteric area)
→I thank you for your very important remarks. I regret that I did not include this point of view in this study. The basic premise of this study was AFF as defined by the ASBMR, and from this point of view, both subtrochanter and femoral diaphysis were included with regard to the fracture site. The results show that, as you pointed out, a large number of subtrochanteric patients were included in the study, and it is undeniable that this may have had a significant influence on the results. This may have been overlooked in the present study, which focused on surgical techniques. We have added this point to the discussion. (Line 311-318)
Does LIPUS use improved union rate?
→Thank you for your very important and critical question. As mentioned in the discussion, we believe that bone formation stimulation therapies such as low-intensity pulsed ultrasound may be effective in the pathogenesis of this fracture. However, even if it can be shown to be effective in general fractures, it remains to be seen whether it is effective in conditions such as the present one.
please provide ethical commite approval in the main text
→I put it in as you indicated. (Line195)
does surgeries outcomes varied when senior surgeon who performed the surgery or resident ?
→Thanks for pointing this out. Although it is not mentioned in the text, all surgeries in this case were performed by senior surgeons, so comparisons cannot be made.
is there a role of direct adjunctive such as bone marrow aspirate or DBM ? , why bone grafting only used in some cases and not for others. what was the criteria to choose this approach ?
→Under current Japanese law, bone marrow aspirate is legally only to be performed in strictly limited facilities. For this reason, it was not performed at our hospital. Also, DBM has only recently become available in Japan and we did not have the idea to use it in this series. However, we received very good suggestions and would like to try it in the future.
Regarding the indications for bone grafting, as mentioned in the last limitation, a large proportion of the decision was made by the surgeon. As you say, it should have been done with strict indications.
line 171 : why dynamization was done after 31 months ! why no treatment was done before !
→Certainly right. The most important reason was that this patient had no symptoms at all and refused to undergo surgery. After persuasion, it was decided to perform the operation at 31 months. This point will be added in the text. (Line231-232)
how authors can explain that drilling was associated with higher rate of non union (while authors mentioned that drilling was done in case of sclerotic bone) which is mean that there is a confounding bias . i believe that the status of sclerotic bone which was the main cause of non union (not the drilling) . what was the confidance interval ? how different variables were elimianted ?
→ I apologise that this is the most difficult and incomprehensible consideration in the analysis of the results of this study. As you have pointed out, the factor that most inhibits bone fusion in this pathology is biological activity. Osteosclerosis may be the most obvious intraoperative finding as a manifestation of this. The status of the healing process is naturally also a major possibility. The surgeon's subjectivity and preconceptions are likely to be involved in this respect. This point will be added during the discussion. (Line 302-308)
The statistics presented here are based on an analysis of the results of the univariate analysis. The 95% confidence intervals for the Union and Nonunion ratios for drilling ranged from 0.228 to 0.772 respectively. Another reviewer pointed out that the statistical methods should be described in more detail, so we will add this to the statistical methods section. (Line201-203)

Reviewer 2 Report
Comments and Suggestions for Authors
suggestions for the introduction section:
- authors should expand on the mechanism of bisphosphonates, detailing how they affect bone resorption and formation, and their impact on bone quality beyond just osteoclast suppression.
- consider discussing the variability in risk of severely suppressed bone turnover and atypical femoral fractures among different types of bisphosphonates and among patient demographics or conditions.
- authors should provide an overview on how glucocorticoids impair fracture healing, or other risk factors for delayed healing and pseudoarthrosis
- integrate a brief review of recent studies on the surgical treatment of atypical/rare femoral fractures, highlighting any gaps in knowledge or conflicting findings
for the methods section:
- typo in "subtrochabter" should be "subtrochanter".
- overuse of abbreviations without first defining them (e.g., DEXA, LIPUS)
- provide detailed explanation of the diagnostic criteria used and clarify the policy for excluding certain cases
- detail on background on the setting of the study, including the hospital's specialization, the typical patient population, and why the institution was uniquely chosen
- Add a detailed explanation of the statistical methods used, including why certain statistical tests were chosen
Author Response
Response to Reviewer 2
Thank you for taking the time in your busy schedule to review my paper. Thank you also for your many helpful and important comments and suggestions. I have prepared a new, highlighted version of the paper based on the comments of another reviewer and am sending it to you. Deletions have been bylined and new additions have been added in red. Please check it.
Also below is my response to your peer review comments. It is possible that there are some errors in my understanding, but I would be grateful if you would not hesitate to point them out and provide me with guidance. Thank you for your continued patience.
- authors should expand on the mechanism of bisphosphonates, detailing how they affect bone resorption and formation, and their impact on bone quality beyond just osteoclast suppression.
→Thanks for your important remarks. Another reviewer also pointed out the need for a detailed explanation regarding the excessive suppression of bone turnover by BP, which has been added along with the basic mechanism of BP. (Line 57-78, 81-94)
- consider discussing the variability in risk of severely suppressed bone turnover and atypical femoral fractures among different types of bisphosphonates and among patient demographics or conditions.
→Thanks for your valuable suggestions. We have added this point to the discussion. (Line 264-277)
- authors should provide an overview on how glucocorticoids impair fracture healing, or other risk factors for delayed healing and pseudoarthrosis
→Thanks for your valuable suggestions. We have added the mechanism of GC on bone metabolism to the discussion. (Line 251-260)
- integrate a brief review of recent studies on the surgical treatment of atypical/rare femoral fractures, highlighting any gaps in knowledge or conflicting findings
→I added your indicated point to the discussion. (Line 331-336)
for the methods section:
- typo in "subtrochabter" should be "subtrochanter".
→Thanks for pointing this out. It has been corrected. (Line 138)
- overuse of abbreviations without first defining them (e.g., DEXA, LIPUS)
→ As you indicated, the less frequent DEXA and LIPUS were listed without abbreviations. (Line 155-156, 172, 199, 210, 211, 339-340, 343-344)
- provide detailed explanation of the diagnostic criteria used and clarify the policy for excluding certain cases
→A detailed explanation is added regarding the diagnostic criteria used. An exclusion policy is also described. (Line 106-120)
- detail on background on the setting of the study, including the hospital's specialization, the typical patient population, and why the institution was uniquely chosen
→The selected hospital is a university hospital that provides advanced medical care and has a very active department of collagen diseases. As a result, many patients with autoimmune diseases come to the hospital and are referred intra-hospital. Therefore, this is a highly skewed group of AFF patients and differs from the multicenter patient population we have investigated in the past (ref. 8). We have added to the text to that effect. (Line 288-292)
- Add a detailed explanation of the statistical methods used, including why certain statistical tests were chosen
→There was an error in the way the statistical technique was described, which has been corrected and added to. (Line 201-203)

Round 2
Reviewer 2 Report
Comments and Suggestions for Authors
Authors have made the changes. The work is now suitable.